# Global computational alignment of tumor and cell line transcriptional profiles

Allison Warren[1], Yejia Chen[1], Andrew Jones[1], Tsukasa Shibue[1], William C. Hahn [1,2,3], Jesse S. Boehm [1], Francisca Vazquez [1], Aviad Tsherniak [1,4] & James M. McFarland [1,4✉]

Cell lines are key tools for preclinical cancer research, but it remains unclear how well they represent patient tumor samples. Direct comparisons of tumor and cell line transcriptional profiles are complicated by several factors, including the variable presence of normal cells in tumor samples. We thus develop an unsupervised alignment method (Celligner) and apply it to integrate several large-scale cell line and tumor RNA-Seq datasets. Although our method aligns the majority of cell lines with tumor samples of the same cancer type, it also reveals large differences in tumor similarity across cell lines. Using this approach, we identify several hundred cell lines from diverse lineages that present a more mesenchymal and undifferentiated transcriptional state and that exhibit distinct chemical and genetic dependencies. Celligner could be used to guide the selection of cell lines that more closely resemble patient tumors and improve the clinical translation of insights gained from cell lines.

[1] Broad Institute of MIT and Harvard, Cambridge, MA, USA. [2] Department of Medical Oncology, Dana Farber Cancer Institute, Boston, MA, USA. [3] Harvard Medical School, Boston, MA, USA. [4] These authors contributed equally: Aviad Tsherniak, James M. McFarland. ✉email: jmmcfarl@broadinstitute.org

Tumor-derived cell line models have been a cornerstone of cancer research for decades. The genomic and molecular features of over a thousand cancer cell line models have now been deeply characterized[1], and recent efforts are systematically mapping their genetic[2–4] and chemical[5] vulnerabilities. These datasets are thus providing new opportunities to identify potential therapeutic targets and connect these vulnerabilities with measurable biomarkers that can be used to develop precision cancer approaches[2,5].

The clinical applicability of results derived from cancer cell lines remains an important question, however, due largely to uncertainty as to how well they represent the biological characteristics and drug responses of patient tumors. Historically derived cell line models likely represent an incomplete sampling of the spectrum of human cancers[6,7]. Many existing models have been propagated for decades in vitro, with factors such as clonal selection, cell culture conditions, and ongoing genomic instability all potentially contributing to systematic differences between cell line models and tumors[7–9]. Furthermore, many cell line models lack detailed clinical annotations. Therefore, it is critically important to better understand the systematic differences between cell lines and tumors to identify which tumor types have existing cell lines that sufficiently recapitulate their biology and which tumor types do not. Such systematic comparisons may ultimately also help reveal whether patient-derived xenografts, genetically engineered mouse models, or organoid cultures are more, less, or equivalently faithful to human tumors than historical cell lines.

Large datasets such as The Cancer Genome Atlas (TCGA)[10] and the Cancer Cell Line Encyclopedia (CCLE)[1] include the multi-omic features of approximately 10,000 primary tumor biopsy samples and more than 1,000 cancer cell lines. While TCGA focuses primarily on primary tumors (as opposed to metastatic or drug-resistant tumors from which certain cell lines may have been derived) it nevertheless provides a powerful opportunity to begin to perform detailed comparisons of tumors and cell lines systematically across many cancer types. By contrast, previous studies have primarily focused on comparing tumors and cell lines within particular cancer types[11–14].

In principle, a global alignment of the datasets would allow for the identification of the best cell line models for a given cancer type, without relying on annotated disease labels. Existing global analyses have mainly compared samples based on their mutation and copy number profiles[15], which are complicated by several factors: a lack of paired normal samples for calling mutations in cell lines, systematic differences in the overall rates of copy number variation and mutations[13,16–19], as well as being limited to known cancer-related lesions.

Comparisons based on information-rich gene expression profiles are a promising alternative[20], given their demonstrated utility for resolving clinically relevant tumor (sub)types[21–25], as well as predicting genetic[2] and chemical vulnerabilities of cancer cells[5,26]. However, a key challenge is that gene expression measurements from bulk tumor biopsy samples are confounded by the presence of stromal and immune cell populations not found in cell lines, often comprising a substantial fraction of the cellular makeup of each sample[27,28]. The presence of these contaminating cells not only compromises the expression profiles by their own gene expression, but also sends heterotypic signals to the tumors cells that ultimately affect the gene expression within the tumor cells[29,30]. Existing approaches for removing the effects of contaminating cells generally require detailed prior knowledge of the expression profiles of each contaminating cell type[31], and do not account for other systematic differences between in vitro and tumor expression profiles. Furthermore, more general batch effect correction methods typically require either pre-existing subtype annotations, or assume the cell line and tumor datasets have the same subtype composition[32].

To address these challenges, we developed Celligner, a method to perform an unsupervised global alignment of large-scale tumor and cell line gene expression datasets. Celligner leverages computational methods recently developed for batch correction of single-cell RNA-Seq data and differential comparisons of high-dimensional data, in order to identify and remove the systematic differences between tumors and cell lines, allowing for direct comparisons of their transcriptional profiles. Notably, Celligner aligns pan-cancer gene expression datasets without the need for any additional information (such as tumor type labels, contaminating cell signatures, or tumor purity estimates).

We apply our method to tumor data from TCGA, TARGET, and Treehouse[33] and cancer cell line data from CCLE and the Cancer Dependency Map[1]. This comparison identifies cell lines that match well to different tumor subtypes, as well as cancer cell lines that are transcriptionally distinct from their annotated primary cancer types.

## Results

**Alignment of tumor and cell line transcriptional profiles.** To illustrate the analytical challenges involved with directly comparing cell line and tumor expression profiles, we first combined several large RNA-Seq gene expression datasets and performed a joint dimensionality reduction analysis. Specifically, we analyzed transcriptional data from 1,249 CCLE cell lines[1], 9,806 TCGA tumor samples, 784 pediatric tumor samples from TARGET, and 1,646 pediatric tumor samples from Treehouse[33]. Although a consistent computational pipeline was used to process all datasets, this analysis revealed a clear separation of cell line and tumor samples (Fig. 1a), as expected. This separation was not addressed by applying simple normalization or batch correction methods such as ComBat[32,34] (Supplementary Fig. 1). This global separation of cell line and tumor samples precludes more detailed assessments of the similarity of samples of different types.

Several features of the problem make the alignment of cell line and tumor expression profiles challenging. First, the degree of tumor purity is highly variable across tumor samples[27,28], and the transcriptional effects of contaminating normal cells are not captured by a single signature shared across tumor samples and types[35]. Secondly, differences in the disease (sub)type composition of the datasets can greatly confound attempts at globally aligning the distributions of tumor and cell line expression profiles. Finally, even without the confounding effects of normal-cell contamination, differences between in vivo and in vitro conditions, as well as technical artifacts, likely give rise to systematic differences in the cancer cells' gene expression profiles[36].

To address these challenges, we developed a multi-step alignment procedure (Fig. 1b; Methods). First, to identify gene expression signatures characterizing recurrent patterns of normal cell contamination in tumor samples, we used contrastive principal component analysis (cPCA), a generalization of PCA that uncovers patterns of correlated variation that are enriched in one dataset relative to another[37]. Importantly, to avoid biases resulting from the differential disease composition between the two datasets, we first performed an unsupervised clustering of each dataset (Supplementary Fig. 1; Methods; Supplementary Data 1), and used cPCA to contrast the intra-cluster covariance structure between the cell line and tumor data. This analysis identified several gene expression signatures with greatly elevated variance across the tumor samples compared to the cell lines (Fig. 1c). Gene set enrichment analysis (GSEA)[38] of these tumor-specific signatures revealed clear enrichment for immune

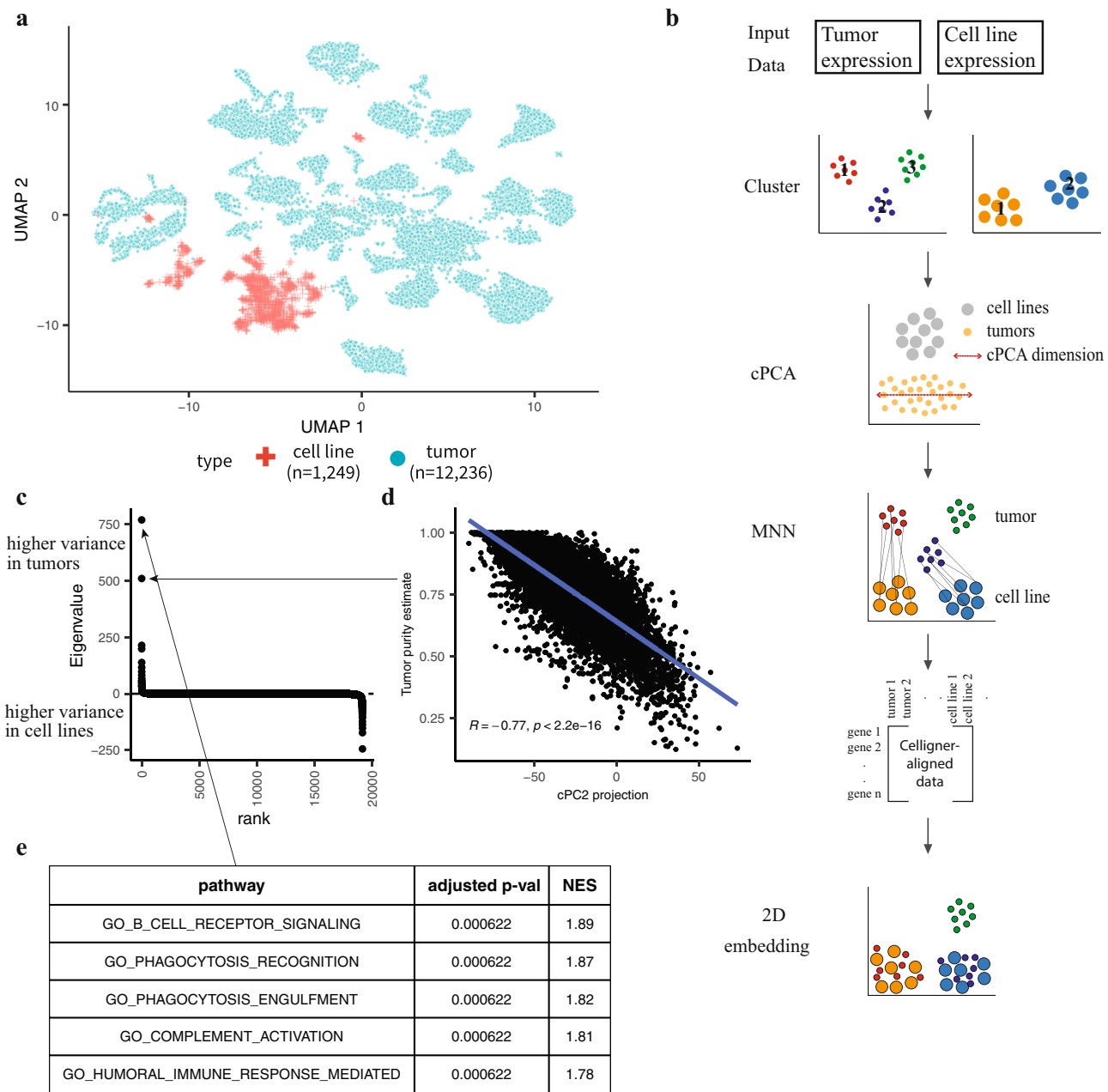

**Fig. 1 Overview of the Celligner alignment method. a** A 2D projection of combined, uncorrected cell line and tumor expression data using UMAP ($n =$ 1,249 cell lines, $n = 12,236$ tumors). **b** Method: Celligner takes cell line and tumor gene expression data as input, and first identifies and removes expression signatures with excess intra-cluster variance in the tumor compared to cell line data using contrastive Principal Component Analysis (cPCA). Then Celligner identifies and aligns similar tumor-cell line pairs to produce corrected gene expression data, using mutual nearest neighbors (MNN) batch correction, which allows for improved comparison of tumors and cell lines. **c** cPC eigenvalues ordered by rank ($n = 19,188$ eigenvalues). **d** Pearson correlation between the projection of tumor samples onto cPC2 and their estimated purity (using a consensus measurement of tumor purity[28]) ($n = 7,832$ tumors). **e** The top five pathways from gene set enrichment analysis (GSEA) of cPC1. P-values are based on a gene-permutation test and adjusted using the Benjamini-Hochberg procedure (Methods, 'Gene set enrichment analysis').

pathways (Fig. 1e; Supplementary Fig. 2), suggesting that cPCA identifies the presence of different contaminating immune cell populations. Furthermore, expression of the second tumor-specific cPC was significantly correlated ($R = -0.77$, *p*-value < 2.2e-16, $n = 7,832$ tumors) to independent estimates of tumor purity based on a consensus measurement of tumor purity[28], illustrating that this analysis is able to identify multiple independent signatures of contaminating cells (Fig. 1d). As the first stage of alignment, we thus removed the top four tumor-specific signatures from both datasets (Methods).

While cPCA removes a dominant source of systematic tumor/ cell line differences, on its own it does not fully align the datasets (Supplementary Fig. 3) as it does not account for uniform differences between tumor and cell line profiles of a given disease (sub)type. As a second stage of the alignment, we utilized a batch effect correction algorithm based on mutual nearest neighbors (MNN) to remove the remaining systematic differences between the datasets. MNN batch correction was developed to remove batch effects in single-cell RNA-Seq data. It functions by identifying pairs of samples between datasets where each sample

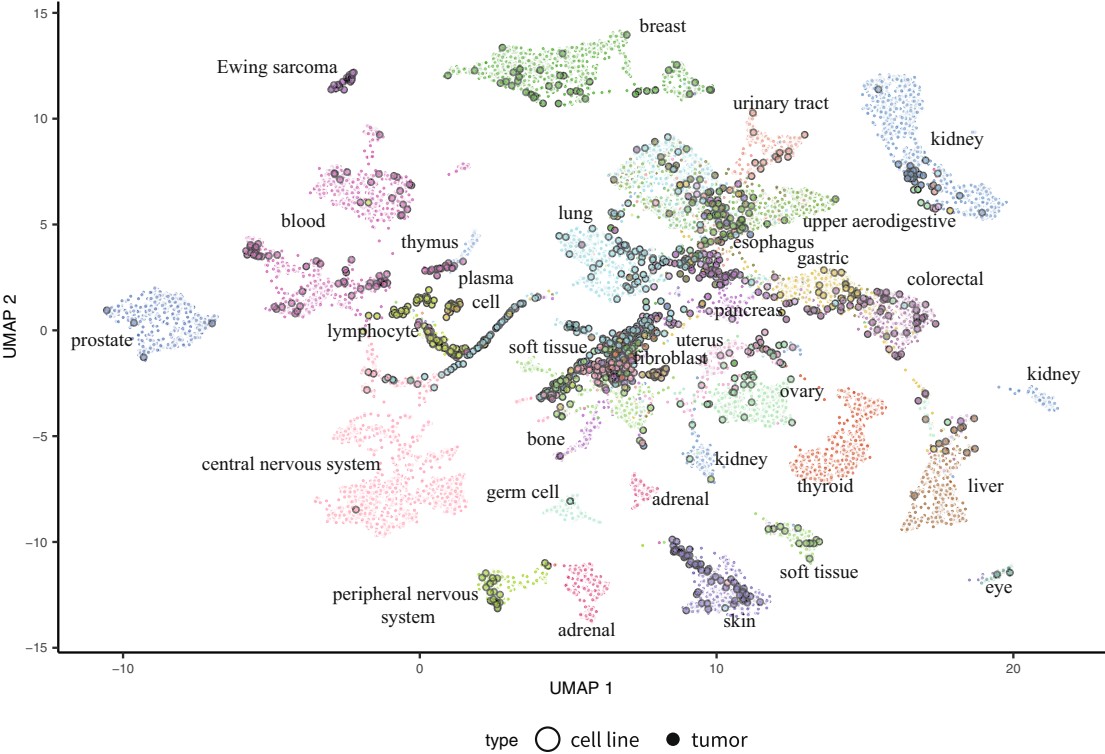

**Fig. 2 Celligner alignment of tumor and cell line samples.** UMAP 2D projection of Celligner- aligned tumor and cell line expression data colored by annotated cancer lineage. The alignment includes 12,236 tumor samples and 1,249 cell lines, across 37 cancer types.

is contained in each other's set of nearest neighbors in the other dataset, and leverages these MNN pairs to learn a flexible but robust nonlinear alignment of the datasets. Critically, MNN is robust to differences in subtype composition between the datasets, assuming only that the datasets contain a subset of corresponding samples[39].

Application of MNN to the tumor and cell line expression profiles identified a set of correction vectors (differences in expression profiles between matched tumor/cell line pairs), which on average showed increased expression of immune-related genes, and decreased expression of cell cycle genes, in tumors compared to MNN-matched cell lines (Supplementary Fig. 3). While these tumor/cell line differences were largely consistent across samples, the correction vectors also showed patterns that varied across disease types (Supplementary Fig. 3), highlighting the importance of using a flexible nonlinear correction method such as MNN to remove such systematic differences. Notably, while MNN on its own provided a broadly similar alignment of the datasets, application of cPCA prior to MNN increased the number of MNN pairs identified, and helped mitigate bias towards matching cell lines with higher-purity tumor samples in MNN pairs (Supplementary Fig. 3).

We applied this two-stage alignment method (which we refer to as Celligner) to produce an integrated dataset of cell line and tumor gene expression profiles that have been corrected for multiple sources of systematic dataset-specific differences. Indeed, creating a 2D Uniform Manifold Approximation and Projection (UMAP)[40] plot with the Celligner-aligned dataset revealed a map of cancer transcriptional profiles with cell line and tumor samples largely intermixed, while still preserving clear differences across known tumor types (Fig. 2).

**Alignment preserves meaningful subtype relationships.** To evaluate Celligner, we first tested whether it produced an alignment of known disease types and subtypes present in both the tumor and cell line data. As apparent in Fig. 2, Celligner removes much of the systematic differences between tumor and cell line expression profiles, producing an integrated map of cancer expression space with clear clusters composed of both cell line and tumor samples. Even though Celligner is completely unsupervised (i.e., does not rely on any sample annotations such as disease type), the aligned tumor and cell line expression profiles largely clustered together by disease type. We quantified this by classifying the most similar tumor type for each cell line, based on its nearest neighbors among the tumor samples. We found that, for disease types found in both datasets, these inferred tumor types matched the annotated cell line disease type 57% of the time (Fig. 3a; Methods), while in the uncorrected data the inferred tumor types matched the annotated cell line disease type 49% of the time (Supplementary Fig. 4). Celligner correction also increased the measured similarity of tumors and cell lines expression profiles of the same type (Fig. 3b; Supplementary Fig. 4).

A key advantage of Celligner is that it does not assume that all cell line samples in a dataset are necessarily similar to any tumor samples, and vice versa. As a result, we can utilize the Celligner-aligned expression data to identify which cancer types show good agreement between cell lines and primary tumors, and which do not. Although a high proportion of cell lines clustered with tumors of the same cancer type, not all cell lines aligned well with tumor samples. For example, while many soft tissue, skin cancer, and breast cancer cell lines were similar to corresponding tumor samples, we found that central nervous system (CNS) and thyroid cell lines consistently aligned poorly with tumor samples (Fig. 3a, b). This observation agrees with previous reports in the literature that in vitro media conditions can alter the phenotype of CNS cell lines and cause genomic changes that were not present in the original tumor[41,42].

Nevertheless, these analyses illustrate that overall, Celligner tends to group cell lines and tumors of the same disease type. We next sought to determine whether the aligned data also reveal

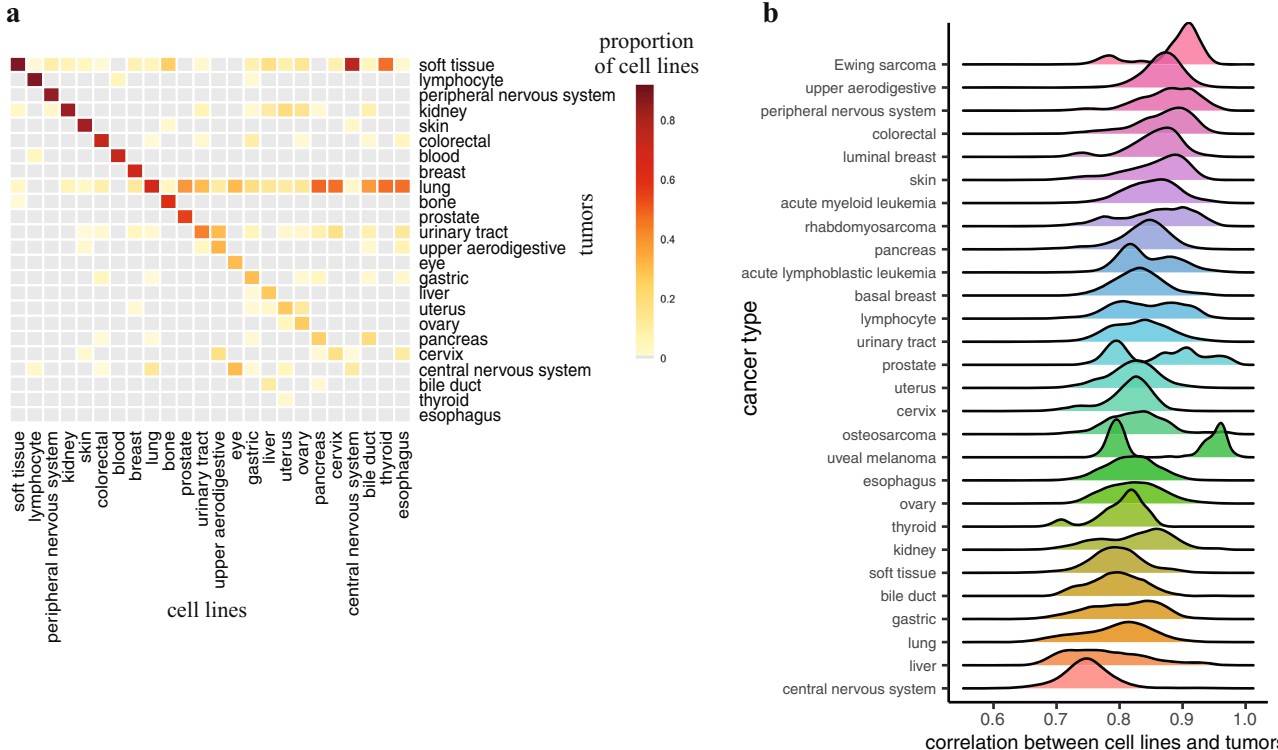

**Fig. 3 Classification of cell lines by tumor type. a** The proportion of cell lines ($n = 1,150$) that are classified as each tumor type using Celligner-aligned data. **b** Distribution of correlations between cell lines ($n = 1,135$) and tumors ($n = 11,413$) of the same (sub)type after Celligner-alignment.

meaningful relationships between more granular subtypes. To this end, we aggregated existing subtype annotations for cell line and tumor datasets (Supplementary Data 1)[1,10,43–45], and found that Celligner also tended to align tumor and cell line samples of the same subtype (Fig. 4a). For example, breast cancer tumors and cell lines clustered together by subtype (Fig. 4b). Similarly, the leukemia samples formed clusters that correspond to existing annotations for acute myeloid leukemia (AML) and acute lymphoblastic leukemia (ALL), and the majority of leukemia cell lines aligned to tumors of the same subtype (Fig. 4c, d).

We further tested whether Celligner preserves and aligns biologically meaningful intra-cluster variability. For example, even though the melanoma tumors and cell lines mainly formed a single distinct cluster, variability within this cluster recapitulated recently-described melanoma differentiation states[43], and annotations of these melanoma subtypes were well-aligned between cell lines and tumors (Fig. 4e). Interestingly, the region of the melanoma cluster that consisted entirely of tumor samples primarily contained tumors of the transitory subtype that are from primary - rather than metastatic - samples. This result is consistent with the fact that many of the melanoma cell lines are annotated as being derived from metastatic samples[1]. Together, these results highlight the ability of Celligner to reveal more detailed patterns of transcriptional similarity between cell lines and tumors, going beyond merely matching clusters.

One potential concern with methods that seek to globally align tumor and cell line data is that they might obscure important underlying biological differences. A key feature of Celligner in this regard is that it allows for sub-populations that are only present in one dataset or the other. For example, both data sets contain renal cancer samples, but samples annotated as chromophobe renal cell carcinoma are only present in the tumor data. Accordingly, after the Celligner alignment, the cluster of chromophobe renal cell carcinoma tumor samples remained distinct and did not include any cell lines (Fig. 4f). Similarly,

myeloma cancer samples are present in the cell line data, but not in the TCGA, TARGET, or Treehouse tumor data. After Celligner correction, the myeloma cell lines clustered near the other hematopoietic samples, but did not clearly group with any tumor samples (Fig. 4d). The cell line data also includes 39 cell lines annotated as fibroblasts, which we expect to be transcriptionally distinct from any cancer type in the tumor or cell line data. Indeed, we found that these fibroblast cell lines formed a distinct cluster with virtually no tumors present (Supplementary Fig. 5). These cases demonstrate that Celligner does not artificially force all samples to align with samples from the other dataset, allowing it to reveal subtypes that are absent or underrepresented in either dataset.

**Celligner enables improved tumor-cell line similarity estimates.** We also assessed the results of Celligner by comparing it to two previously published methods designed to measure the transcriptional similarity of cell lines and tumors. Firstly, the method of Yu et al.[20] used a combination of batch effect correction (using ComBat[32]) and regression-based tumor-purity correction to calculate adjusted similarity metrics between cell lines and tumors. We also compared Celligner to the results of CancerCellNet[46], a method that uses a machine learning model trained on tumor transcriptional profiles to predict the most likely tumor type of each cell line.

Overall, we found that estimates of which cell lines were more or less transcriptionally similar to tumors of the same annotated type were highly concordant between Celligner and both the method of Yu et al.[20] (Supplementary Fig. 6), and CancerCell-Net[46] (Supplementary Fig. 7; Methods). We further compared the ability of each of these methods to accurately classify the annotated cancer type of cell lines based on their similarity to tumor profiles. For Celligner and the Yu et al.[20] method we based these classifications on each cell line's nearest neighbors (using Pearson correlation; Methods), while for CancerCellNet[46] we

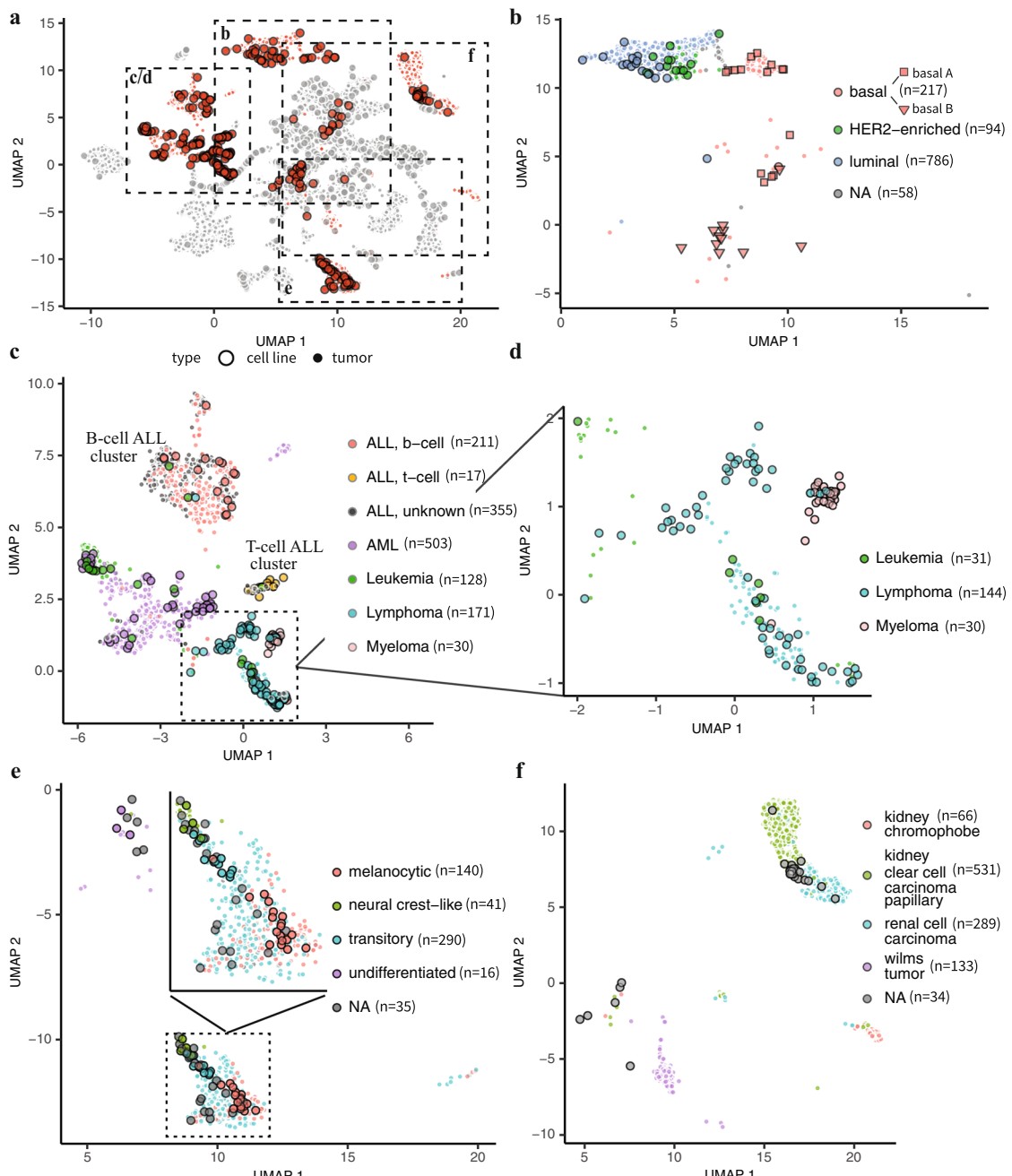

**Fig. 4 Subtype alignment. a** UMAP 2D projection of the Celligner-alignment with breast, kidney, hematopoietic, and skin samples highlighted in red.
**b** Luminal and basal breast cancer subtypes cluster together for cell lines and tumors ($n = 56$ cell lines, $n = 1,099$ tumors). **c** Leukemia tumor and cell lines
co-cluster in the aligned data by subtype ($n = 197$ cell lines, $n = 1,218$ tumors). **d** Myeloma cell lines cluster near the hematopoietic samples, but distinct
from the tumors ($n = 96$ cell lines, $n = 109$ tumors). **e** Skin cancer melanocytic, transitory, neural crest-like, and undifferentiated subtypes align between
cell lines and tumors ($n = 63$ cell lines, $n = 459$ tumors). **f** Kidney chromophobe tumor samples do not cluster with any cell lines ($n = 34$ cell lines, $n = 1019$
tumors).

used the tumor type probability estimates for each cell line. This
comparison showed that Celligner-based classifications were
substantially more accurate compared to those derived using
the method of Yu et al.[20] (Celligner: 50% agreement, Yu et al.:
38% agreement; $n = 666$ cell lines, $n = 7,656$ tumors), as well as
when compared to the estimates from CancerCellNet[46] on
matched cohorts (Celligner: 49% agreement, CancerCellNet:
37% agreement; $n = 657$ cell lines, $n = 8,825$ tumors; **Methods**).

Lastly, we sought to compare Celligner, which uses transcrip-
tional data to compare cell lines and tumors, with similarity
metrics based on genomic features. There are a large number of

potential genomic features and representations that could be used
for such an analysis. To simplify the comparison, we utilized a
previously published set of 1,250 binarized cancer functional
events (CFEs)[15,47], which include copy number, methylation, and
mutation features identified as recurrent events in tumors
(Methods). We found that both Celligner-based and CFE-based
similarity estimates produced similar accuracy when classifying
the cell lines' cancer type based on tumor data (Celligner: 60%
agreement, CFEs: 61%; $n = 2,448$ tumors, $n = 459$ cell lines;
Methods). However, the rankings of which cell lines were most
similar to tumors of the same type differed substantially

(Supplementary Fig. 8), suggesting that transcriptomic and genomic similarity may vary independently.

**Information transfer between cell line and tumor datasets**. By providing an unsupervised data integration procedure, Celligner enables joint analyses of the tumor and cell line datasets, providing greater power to detect transcriptionally distinct subpopulations. This is particularly true for the cell line data where there are ~10-fold fewer samples compared to the tumors. Indeed, clustering analysis of the Celligner-aligned dataset revealed a larger number of more distinct clusters among the cell lines compared with the same analysis applied to the cell line dataset on its own (Supplementary Fig. 1). This difference was most evident for cancer types that had few representative cell lines. For example, in the current dataset, only one testicular cell line is present (SUSA), and when analyzing the cell line data on its own, this cell line clustered most closely with the soft tissue cancers (Supplementary Fig. 9). Joint analysis of the Celligner-aligned data, however, showed that this cell line clustered with a subset of the germ cell tumors (Supplementary Fig. 9), and in particular was nearest neighbors with the non-seminoma testicular cancer samples (Supplementary Fig. 9).

We next explored whether integrated analysis of the tumor and cell line data might also help resolve missing, or potentially incorrect (sub)type annotations. For example, four cell lines that are not annotated as melanoma samples nevertheless clustered with the melanoma samples (Supplementary Fig. 9). One such cell line, COLO699, is annotated as derived from a metastatic lung cancer sample[1], raising the possibility that the current annotation accurately characterizes the biopsy site, but not the primary tissue. Previous reports in the literature have also identified that this cell line likely derives from a melanoma sample[11].

We can also use the combined dataset to perform label transfer of annotations from one dataset to another. For example, ALL subtype annotations (T-cell and B-cell) were available for the ALL cell lines, but only for some of the ALL tumor samples. The ALL cell lines formed two distinct clusters, which perfectly matched the labeled B-cell and T-cell subtype. The ALL tumors also largely clustered together with the ALL cell lines, with all of the annotated (B-cell) tumor samples clustering with the B-cell cell lines. The rest of the (un-annotated) tumor samples could easily be classified as either B-cell or T-cell ALL (with some putative AML samples as well) based on their cluster membership (Fig. 4c), which aligned well with the clustering of the tumor samples based on the expression of B-cell ALL and T-cell ALL marker genes (Supplementary Fig. 9)[48]. These results further highlight the advantage of performing an unsupervised global alignment that does not rely on existing annotations.

**A group of transcriptionally and functionally distinct cell lines**. Jointly analyzing the Celligner-aligned cell line and tumor data also revealed common structures across cell lines that were distinct from primary tumors. As described above (Fig. 3), cell line models of certain cancer types, such as thyroid and CNS, did not recapitulate the disease-specific transcriptional patterns exhibited by primary tumors of their respective cancer types. Closer inspection revealed that 252 of the cell lines that did not group with tumor samples of the same disease type formed a separate cluster (Fig. 5a; Methods). While approximately 20% of the cell lines belonged to this cluster, it contained less than 2% of the tumor samples (primarily soft tissue and bone tumors) (Supplementary Data 1). The cell lines in the cluster spun a wide range of different lineages (notably, 82% of all CNS, 91% of all thyroid lines, and 41% of all liver lines; Fig. 5b; Supplementary Data 1),

and we did not identify any distinguishing features based on available clinical annotations for these cell lines (Supplementary Fig. 10).

Cell lines in this cluster lacked lineage-specific expression characteristics present in the primary tumor datasets analyzed herein, suggesting that they were derived from an undifferentiated tumor or have entered a more undifferentiated state. Indeed, of the twelve skin cancer samples in this cluster that were annotated by Tsoi et al., all three of the skin cell lines and seven of the nine skin tumors were annotated as being of an undifferentiated subtype[43]. The majority (11/12) of the thyroid cell lines, which have been observed to be more dedifferentiated than thyroid tumors[36,49,50], also belonged to this cluster. To further assess how distinct these cell line models were from their lineage-matched counterparts that co-clustered with tumors, we also looked at a set of lineage-specific transcription factors. For example, SOX10 is selectively expressed in melanoma cells, and SOX10 knockout by CRISPR is lethal selectively in melanoma cell lines[3]. Consistent with the interpretation of this cell-line-specific cluster as representing a more dedifferentiated state, skin cancer lines within the cluster showed much weaker expression of, and dependency on, SOX10 (Fig. 5c)[43]. Similarly, liver cancer cell lines within the undifferentiated cluster showed lower expression of, and less dependency on, the hepatocyte transcription factor HNF4A (Fig. 5d).

To further understand the biological features that distinguish this group of undifferentiated cell lines we performed genome-wide differential expression analysis, controlling for differences attributable to the annotated lineages. This analysis revealed a striking enrichment of epithelial-mesenchymal transformation (EMT)-related genes (Fig. 5e, f), reflecting a stronger mesenchymal expression pattern among these undifferentiated cell lines. The relatively small set of tumor samples in this cluster ($n = 229$) were primarily from cancer types with mesenchymal cell lineages[51], and GSEA showed that these samples exhibited elevated expression of genes in the EMT pathway (normalized enrichment score = 3.33, adjusted p-value = 6.2e-05). Notably, however, there were a small number of tumor samples from other lineages present in the cluster (Supplementary Data 1), including all of the melanoma tumors annotated as the undifferentiated subtype (Fig. 4e)[43].

We also tested whether the cell lines expressing this distinct mesenchymal/undifferentiated expression pattern exhibit a unique pattern of chemical and genetic vulnerabilities. For this, we used the Achilles dataset of genome-wide CRISPR knockout screens to interrogate gene essentiality across 689 cell lines[52], as well as a recently-generated dataset of clinical compounds screened across 578 cell lines[5]. These analyses showed that the undifferentiated cell lines have increased sensitivity to tubulin polymerization inhibitors (Fig. 5g), as well as greater dependency on integrin genes, particularly ITGAV and ITGB5 (Fig. 5h), consistent with their upregulation of EMT-related genes[53,54]. The undifferentiated cell lines were also more resistant than other cell lines to several compounds, most notably many EGFR inhibitors (Fig. 5g). This is also consistent with a marked decrease in EGFR dependency among the undifferentiated cell lines (Fig. 5h).

## Discussion
Cancer cell lines are crucial drivers of preclinical cancer research. Yet, our limited understanding of the similarities and differences between cell lines and patient tumors remains a key challenge for translating findings from cell lines to the clinic. To help address this, we developed a computational method, Celligner, which identifies and removes systematic differences between gene expression profiles of tumors and cell lines in an unsupervised

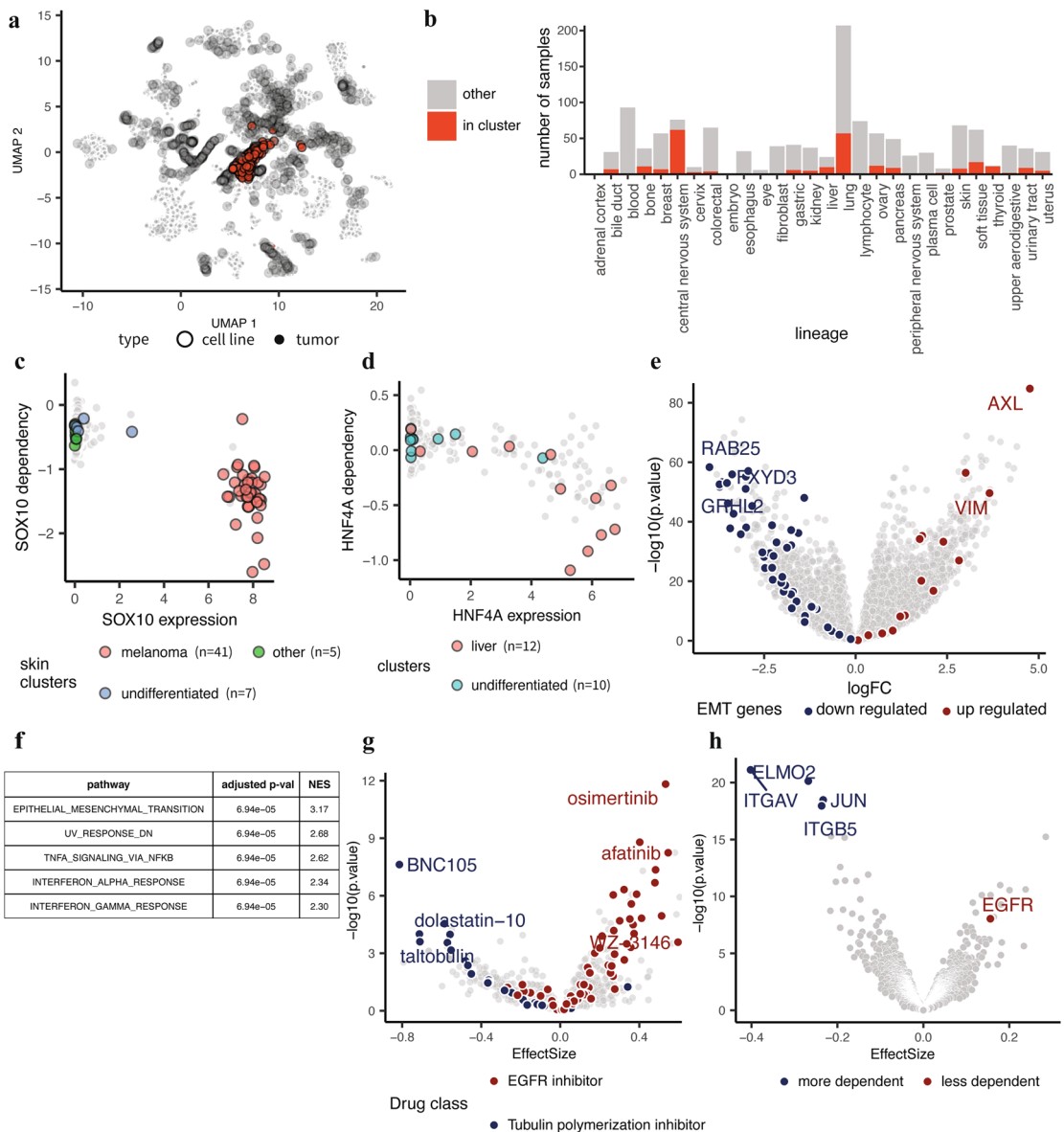

**Fig. 5 A cluster of cell lines show EMT signature and integrin-related dependencies. a** A cluster of 252 undifferentiated cell lines within the global Celligner-alignment. **b** Composition of the cell lines within the cluster (n = 252 cell lines). **c** Skin cell lines within the cluster do not express and do not depend on SOX10. **d** Liver cell lines within the cluster have lower expression of, and weaker dependency on, HNF4A. **e** Differential expression analysis shows an up-regulated mesenchymal profile and **f** enrichment of the EMT pathway for cell lines in the cluster. P-values are based on a gene-permutation test and adjusted using the Benjamini-Hochberg procedure (Methods, 'Gene set enrichment analysis'). **g** Differential drug vulnerability analysis shows decreased sensitivity to EGFR inhibitors and increased sensitivity to tubulin polymerization inhibitors for cell lines in the cluster. **h** Differential dependency analysis shows stronger integrin-related dependencies for cell lines within the cluster.

manner, allowing for direct and detailed comparisons of the transcriptional states of cell lines and tumors.

In our global analysis of 1,249 cell lines and 12,236 tumors, we identified pronounced differences across cancer types in how well cancer cell lines reflected the transcriptional patterns of their primary tumor counterparts. While many disease types (such as lymphoma, Ewing sarcoma, and melanoma) were similar between cell lines and tumors, there were few thyroid and CNS cell lines whose gene expression profiles aligned with the corresponding primary tumor samples. Previous studies have identified that CNS cell lines grown in serum-containing media tend to lose their ability to differentiate, and have gene expression profiles that are unlike their primary tumors[42,55], while CNS cell lines grown in serum-free specialized media had gene expression profiles and genetic aberrations that better recapitulated their primary

tumors[55]. We note that even cell lines that are transcriptionally distinct from their corresponding tumors likely reflect specific dependencies of patient tumors, such as PDGFR dependency, which is found in GBM cell lines and recurrently amplified in tumors[56], and thus can still serve as valuable cancer models. Nevertheless, the tumor-type specific differences revealed in our analysis pinpoint where new cell lines, organoid models[57], patient-derived xenografts, and mouse models are most needed. They also reinforce the importance of efforts such as the Human Cancer Models Initiative (HCMI) that aim to address gaps in our current in vitro model representation. Future applications of the method to RNAseq datasets from these and other novel model formats should prove useful.

Using Celligner, we discovered a distinct set of cell lines, composed of a range of tissue types, which exhibited a

transcriptional state that was largely dissimilar from those of the available primary tumor samples. These cell lines had undifferentiated characteristics, lacking activity of lineage-specific transcription factors (and associated genetic dependencies). They also showed clear upregulation of mesenchymal genes and downregulation of epithelial markers; all characteristics concordant with an EMT phenotype. Consistently, this group of cell lines was most similar to tumors that arise from mesenchymal tissue, but generally clustered separately from the primary tumor samples. Interestingly, these undifferentiated cell lines also exhibited distinct genetic and chemical vulnerabilities, including increased dependency on integrin genes and sensitivity to tubulin inhibitors, as well as decreased sensitivity to EGFR inhibitors, all of which are consistent with an EMT state[58–60]. This group of cell lines included those previously annotated as undifferentiated melanoma and basal B breast cancer subtypes (Fig. 4b), both known to exhibit more stem-like and mesenchymal expression profiles, and associated with more invasive and therapy-resistant cancers[43,61–63]. This raises the possibility that these cell lines may reflect a biologically relevant tumor cell state that is not represented in the primary tumor datasets used here. Indeed, there were a small subset of tumor samples from diverse lineages that clustered with the undifferentiated cell lines, including all of the melanoma tumors previously annotated as undifferentiated[43]. Consistent with this possibility, a stem-like and mesenchymal expression program has been previously identified specifically in early metastatic samples[64], while the tumor data we used is largely from primary tumor samples[10]. Furthermore, EMT gene expression patterns may be obscured in bulk tumor data, as single-cell RNA-Seq analysis of tumors has shown that EMT programs are activated in a minority of cells[65,66]. More research is needed to determine whether these cell lines could be good models for particular tumor cell states, or if they reflect an artifact of cell culture conditions. As new large-scale datasets of metastatic and drug-resistant tumors emerge we can incorporate them into Celligner to better answer this question.

Our analyses focused on using gene expression data to compare tumor and cell line samples. In contrast, previous efforts, such as CELLector[15], have utilized genomic alterations to identify cell lines that are most representative of specific disease subtypes. When we compared the estimates of tumor/cell line similarity produced by Celligner to those based on a set of 1,250 curated copy number, methylation, and mutation features[47], we found that they were largely dissimilar, though both approaches yielded similar accuracy at classifying cell lines' tumor type. We also found that some cell lines previously identified as poor models based on copy number and mutations were identified as non-tumor-like based on our analysis of Celligner-aligned gene expression features as well. For example, Domcke et al. observed that OC316 was hyper-mutated[12], Sinha et al. found that SLR20 had an outlier copy number profile[67], and Ronen et al. found that COLO320 was dissimilar to colorectal tumors and lacked major colorectal cancer driver genes[68]. In our analysis, all of these cell lines were also identified as being unlike their respective tumor types. Overall, these analyses suggest that transcriptional and genomic similarity estimates could reflect distinct aspects of the biology, and might provide complementary information, though further work will be needed to understand these relationships in detail. Indeed, a future version of Celligner that also integrates genomic features could enable more detailed comparisons of tumors and cell lines.

A key component of Celligner is correcting for systematic differences between tumor and cell line expression profiles, most notably those related to the presence of normal cells in tumor samples. To do this, we utilized an unsupervised approach that did not depend on predefined signatures of the various

contaminating cell types, and that also allowed us to account for unknown systematic differences between tumors and cell lines. For instance, we found that cell lines exhibited upregulation of cell cycle expression programs compared with tumors (Supplementary Fig. 3), which agrees with previous findings that a higher proportion of cancer cells are cycling in vitro compared to in vivo[69]. The tumor/cell line differences we identified also varied across disease types, emphasizing the importance of using a nonlinear method that allows for disease-type-specific differences. As single-cell data from normal tissues become more readily available, methods that use these data to estimate and remove the effect of different non-cancerous cells[70,71] could be incorporated to further improve comparisons between tumors and cell lines.

In order to facilitate the use of Celligner, we have incorporated an interactive web app on the Cancer Dependency Map portal (https://depmap.org/portal/celligner), that allows users to explore a Celligner-aligned integrated resource of cell line and tumor expression profiles, as well as download the data. This tool enables the identification of cell line models that best represent the transcriptional features of a tumor type, or even a particular tumor sample, of interest. We hope that an improved understanding of the similarities between cancer cell lines and tumors will allow for better selection of models and allow for better translation of findings on drug response from preclinical models to clinical samples[72]. More generally, by identifying and removing many of the confounding differences between cell lines and tumors in an unbiased fashion, Celligner enables integrated analyses of cell line and tumor datasets that can be used to reveal patterns within, and relationships between, these data, helping to improve translation of insights derived from cell line models to the clinic.

## Methods

**Expression data**. Gene expression data for 12,236 tumor samples were taken from the Treehouse Tumor Compendium V10 Public PolyA dataset, obtained from Xena browser (https://xenabrowser.net)[33] and produced by the Treehouse Childhood Cancer Initiative at the UC Santa Cruz Genomics Institute. The data set compiled samples from the UCSC Treehouse Childhood Cancer Initiative, the Therapeutically Applicable Research to Generate Effective Treatments (TARGET) program, and The Cancer Genome Atlas (TCGA)[10]. Cell line gene expression data for 1,249 samples were taken from the DepMap Public 19Q4 file: CCLE_expression_full.csv[52]. All gene expression data were processed using the STAR-RSEM pipeline and are TPM $\log_2$ transformed (with a pseudocount of 1 added). Gene expression data were subset to 19,188 protein-coding genes that were present in both the tumor and cell line data.

**Celligner method**. To remove sources of variation that are unique to one of the data sets and align the cell line and tumor data we used a multi-step process. First, we used contrastive principal component analysis (cPCA)[37] to identify correlated variability that is enriched in the tumor data compared to the cell line data, or vice-versa. In order to avoid identifying signatures related to differences in the cancer type or subtype compositions of the datasets we first clustered the tumor and cell line data separately and subtracted the average expression of each cluster from all samples in the cluster to estimate the average intra-cluster covariance for tumors and cell lines. The data sets were clustered in 70-dimensional PCA space using a shared nearest neighbor (SNN) based clustering method implemented in the Seurat R package[73], with a resolution parameter of 5. We then regressed out the first four cPCs (components which had higher variance in the tumor data) from both the tumor and cell line data (Supplementary Fig. 11).

We then performed mutual nearest neighbors (MNN) correction[39] on the data sets, using the cell line data as the reference dataset. To identify mutual nearest neighbors between the two datasets we used a set of genes that showed high between-cluster variance in each data set. Specifically, we used limma[74] to estimate the across-cluster variation in each gene's expression within each dataset, using the empirical-Bayes moderated F-statistics as a metric of between-cluster variability. We used the union of the top 1000 genes from each data set with the highest F-statistics (Supplementary Data 2). We modified the MNN algorithm from the R package scran[75] to use different $k$ values (the numbers of nearest neighbors to consider) for each data set, which was necessary to account for the much larger set of tumor samples used compared with cell lines. Specifically, we used a $k$ value of 5 to identify nearest neighbors in the cell line data and a $k$ value of 50 to identify nearest neighbors in the tumor data. We verified that the output was robust to

modest changes in these parameters (Supplementary Fig. 11) and stable even if a tissue type was removed from one of the datasets (Supplementary Fig. 11).

**Measuring tumor/cell line similarity**. To evaluate the similarity of cell lines to tumor samples we used the Pearson correlation distance between each cell line and tumor in the Celligner-aligned space. Cell lines were classified as a tumor type by identifying the most frequently occurring tumor type within each cell line's 25 highest correlated tumor neighbors.

**Yu et al. comparison**. To calculate tumor type classifications for each cell line we used the pairwise correlation matrix provided by Yu et al.[20] and used the approach described above ('Measuring tumor/cell line similarity') to classify each cell line as a tumor type. To compare to the Celligner results we re-classified each cell line using the same approach, but this time using the same subset of cell lines and tumors, as well as the disease categories, defined by Yu et al. To evaluate agreement with annotated types we used the annotations from Yu et al. and only considered cell lines where the annotated type was present within the set of tumors ($n = 666$ cell lines).

**CancerCellNet comparison**. To calculate tumor type classifications for each cell line we used the random forest probabilities output by CancerCellNet and used the maximum probability to classify each cell line as one of the 22 cancer types. To compare to the Celligner results we re-classified each cell line by identifying the tumor type most frequently occurring within each of the cell line's 25 high correlated tumor neighbors (using Pearson correlation within the Celligner-aligned data), but this time using the same subset of cell lines and tumors, as well as the disease categories, defined by Peng et al.[46] To evaluate agreement with annotated types we used the annotations from Peng et al. and only considered cell lines where the annotated type was present within the set of tumors (n = 657 cell lines).

**Cancer Functional Event comparison**. To compare Celligner results to comparisons of cell lines and tumors using the CFE data we calculated tumor type classifications for each cell line using by calculating Jaccard similarity between cell lines and tumors in the binary CFE matrix. Each cell line was classified as the majority tumor type within its 25 nearest neighbors (same approach as described above). We only used samples that were present in both the Celligner and CFE data (n = 2,448 tumors). To compare to the Celligner results we re-classified each cell line using the same approach described above ('Measuring tumor/cell line similarity'), but this time only using cell lines and tumors included within the Iorio et al. dataset[47]. To evaluate agreement with annotated types we used the lineage annotations (Supplementary Data 1) and only considered cell lines where the annotated type was present within the set of tumors (n = 459 cell lines).

**Undifferentiated cluster**. The undifferentiated cluster was identified using a shared nearest neighbor (SNN) based clustering method implemented in the Seurat R package in the 70-PCA space. Two neighboring clusters were combined to form the undifferentiated cluster, as both clusters were composed of a high proportion of cell lines (compared to tumors) and cell lines within both clusters had similar up-regulated mesenchymal expression profiles.

**Differential expression analysis**. Differential expression analysis was performed on gene-level read count data using the 'limma-trend' pipeline[74,76]. We first subsetted the data to genes that had a counts-per-million value greater than one in 10 or more samples. The data were normalized per sample using the 'TMM' method from the edgeR package[77], and transformed to log2 counts-per-million using the edgeR function 'cpm'. Linear model analyses, with empirical-Bayes moderated estimates of standard error, were then used to identify genes whose expression was most associated with covariates of interest, such as disease type, or membership in a particular cluster. When analyzing differential gene expression related to the 'undifferentiated' cell line cluster (Fig. 5e), we included disease type as a covariate in the model. The differential dependency analysis and differential drug analysis were also performed using the limma pipeline[74,76] with empirical Bayes moderated t-stats for p-values and disease type included as a covariate.

**Dependency data**. We used estimates of gene dependency taken from the Achilles genome-wide CRISPR-Cas9 KO data[3], 19Q4 release[1]. Specifically, we used gene effect estimates based on the CERES algorithm, taken from the file DepMap Public 19Q4 Achilles_gene_effect.csv[52].

**Drug sensitivity data**. Cell line drug sensitivity data were taken from a dataset of repurposing drugs screened with PRISM[5]. For the PRISM dataset replicate-collapsed, log fold change data at a 2.5 μM dose from the secondary screen were used. Specifically, we used the 'secondary-screen-replicate-collapsed-logfold-change' and 'secondary-screen-replicate-treatment-info'[78]. Annotations of compound mechanism of action (MOA) were also taken from 'repurposing related drug annotations' from the CLUE data library (clue.io/data).

**Gene set enrichment analysis**. For gene set enrichment analysis of gene expression profiles we used the fgsea R package[79]. We used gene-level statistics (log fold change values in Fig. 5d, contrastive principal component loadings in Fig. 1e, Supplementary Fig. 2, and average MNN correction vectors in Supplementary Fig. 3), and 100,000 permutations of the gene-level values to calculate normalized enrichment scores and statistical significance for gene sets from the 'Hallmark' and 'GO_biological_proccesses' gene set collections from MSigDB v6.2[80]. GSEA results shown display the enrichment score normalized to mean enrichment of random samples of the same size and the associated Benjamini-Hochberg adjusted p-values.

**2D embedding**. To compute 2D embeddings of gene expression profiles (e.g. Figure 2; Supplementary Data 1) we used the UMAP method[40], as implemented in the Seurat v3 package[73]. The UMAP embedding was computed on the first 70 principal components, using Euclidean distance, with an 'n.neighbors' parameter of 10, and a 'min.dist' parameter of 0.5.

**Reporting summary**. Further information on research design is available in the Nature Research Reporting Summary linked to this article.

## Data availability

All datasets used in the analysis are publicly available. The inputs to the Celligner method are RNA-Seq data from TCGA (https://www.cancer.gov/about-nci/organization/ccg/research/structural-genomics/tcga), TARGET (https://ocg.cancer.gov/programs/target), Treehouse (https://treehousegenomics.soe.ucsc.edu/public-data/), and CCLE (https://depmap.org/portal/ccle). The results shown here are in part based upon data generated by the TCGA Research Network: https://www.cancer.gov/tcga[10]. The combined tumor data (TCGA, TARGET, and Treehouse) are available from Xena Browser[33] (https://xenabrowser.net/datapages/?dataset=TumorCompendium_v10_PolyA_hugo_log2tpm_58581genes_2019-07-25.tsv&host=https%3A%2F%2Fxena.treehouse.gi.ucsc.edu%3A443&removeHub=http%3A%2F%2F127.0.0.1%3A7222). CCLE RNA-Seq data, as well as additional cell line data used in analyses in this study, are available from figshare, https://doi.org/10.6084/m9.figshare.11384241.v2[52]. Additional cell line drug screening data used in analyses in the study are also available from figshare, https://doi.org/10.6084/m9.figshare.9393293.v4[78]. The results of running the Celligner method on these datasets are provided on figshare, https://doi.org/10.6084/m9.figshare.11965269.v4[81]. The remaining data are available within the Article, Supplementary Information, or available from the authors upon request.

## Code availability

The full source code implementing the method and generating figures is made available at [https://github.com/broadinstitute/Celligner_ms][82], https://doi.org/10.5281/zenodo.4162468. All R packages used in the method and to generate figures are included in the script, install_packages.R, within the repo.

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

## Acknowledgments
This work was supported in part by NCI U01 CA176058 (W.C.H.) and NCI Cancer Model Development Center [Task Order Number HHSN26100008 under Contract no. HHSN261201500003I] (J.S.B.).

## Author contributions
A.W., A.T., and J.M.M. conceived and designed the study. A.W. and J.M.M. wrote the analysis code. A.W. performed the analysis with help from A.J., Y.C. created the online interactive tool. T.S., F.V., J.S.B. helped with interpretation. A.W. wrote the manuscript. W.C.H., J.S.B., F.V., A.T. and J.M.M. reviewed and revised the manuscript. A.T. and J.M.M. supervised the study. W.C.H., J.S.B., and F.V. acquired funds for the study.

## Competing interests
W.C.H. is a consultant for ThermoFisher, Solasta, MPM Capital, iTeos, Frontier Medicines, and Paraxel and is a Scientific Founder and serves on the Scientific Advisory Board (SAB) for KSQ Therapeutics. F.V. receives research support from Novo Ventures. A.T. is a consultant for Tango Therapeutics. All authors were partially funded by the Cancer Dependency Map Consortium, but no consortium member was involved in or influenced this study.
