## [Peer Review File · Nature Communications]

Reviewers' Comments:

Reviewer #1:

Remarks to the Author:

The authors tackle an important problem comparing cell lines to tumor samples leveraging global transcriptional profiles. They developed an unsupervised alignment method (Celligner) and applied it to integrate several large-scale cell line and tumor RNA-Seq datasets. The method aligns the majority of cell lines with tumor samples of the same cancer type and reveals large differences in tumor/cell line similarity across disease types. Furthermore, Celligner identifies a distinct group of several hundred cell lines from diverse lineages that present a more mesenchymal and undifferentiated transcriptional state and which exhibit distinct chemical and genetic dependencies. The manuscript is well written and is easy to follow. The authors aggregate several very large datasets and the analysis is robust and innovative however several points need to be addressed prior to publication:

- The version of the manuscript that was submitted seems to have track changes on
- Figure 1a, it would be useful to assign the shape to the dots of the samples per study
- The authors note the importance of tumor purity – one additional step in the analysis could be to account for tumor purity in their analysis
- Figure b – what is the motivation behind using top 70 PCs, has a sensitivity analysis been done?
- A comprehensive comparison to the rankings based on simple correlation from Yu et al and Collector would be helpful to evaluate the approach.
- Have the authors considered other validation approaches?
- Integrating mutation data to see if similar cell lines are closer to tumor types of interest based on various omics measurements would be worthwhile.
- Are there any clinical or other annotation features of the set of “novel group of transcriptionally and functionally distinct cell lines”? Maybe they are older?

Minor:

Page 2 typo: tumora should be “tumors”

Reviewer #2:

Remarks to the Author:

Overall, this is a well-written and timely paper about the important bioinformatic problem of matching and assessing the relevance of cell lines to the tumor tissues from which they were derived. The authors adapt data harmonization methods commonly used in single-cell RNA-seq for matching cell types across disparate data sets to great effect at the level of gene expression, and even demonstrate tumor subtype-level matching and point out systematic issues with, for example, cell lines derived from CNS tumors. I am enthusiastic about this paper, but have the following suggestions:

1) It would be interesting to include a real "negative control" cell line or cell type in this analysis. The authors show important trends like the overall poor alignment of CNS- and thyroid-derived cell lines with tumor tissue expression profiles, but it would be worth knowing how a completely non-neoplastic cell aligns to the tumor tissues. One could even take something like a relatively homogeneous cardiac-derived cell (a tissue where tumors don't really form).

2) While the methodology is well-described here, the study lacks a systematic performance comparison to competing methods, which I think should be an essential part of any paper primarily reporting a new technique. The authors mention the work of Yu et al, Nature Communications, 2019 which used Combat and a specific correlation-based strategy to compare cell lines and tumor tissues. While the authors show a UMAP projection of COMBAT results in Supp. Fig. 1, there is a little more to the particular methodology of Yu et al, and there's no real quantitative comparison - just a visualization. In addition, there are other prominent studies that developed very different approaches to this problem. For example, Alvarez et al, Nature Genetics, 2019 also tackles this problem of

matching cell lines and other patient-derived models to tumor tissue at the level of gene expression. They use protein activities inferred from RNA-seq data via regulatory network reconstruction. While these are just suggested points of comparison, I feel that some type of quantitative assessment of how this new approach performs relative to others would improve the paper.

Detailed responses to reviewers

Reviewer #1:

The authors tackle an important problem comparing cell lines to tumor samples leveraging global transcriptional profiles. They developed an unsupervised alignment method (Celligner) and applied it to integrate several large-scale cell line and tumor RNA-Seq datasets. The method aligns the majority of cell lines with tumor samples of the same cancer type and reveals large differences in tumor/cell line similarity across disease types. Furthermore, Celligner identifies a distinct group of several hundred cell lines from diverse lineages that present a more mesenchymal and undifferentiated transcriptional state and which exhibit distinct chemical and genetic dependencies. The manuscript is well written and is easy to follow. The authors aggregate several very large datasets and the analysis is robust and innovative however several points need to be addressed prior to publication:

We thank the reviewer for the positive comments.

- The version of the manuscript that was submitted seems to have track changes on

We apologize for that error and have updated the manuscript so that all past changes have been accepted and track changes just reflect new additions.

- Figure 1a, it would be useful to assign the shape to the dots of the samples per study

We have changed the shape of the cell line samples to improve the clarity of the plot and further distinguish them from the tumor samples (revised Figure 1a).

- The authors note the importance of tumor purity – one additional step in the analysis could be to account for tumor purity in their analysis

We thank the reviewer for bringing up this point. Indeed, we decided to use unsupervised methods to correct for the effects of normal cell contamination of tumor samples, which do not rely on pre-existing estimates of tumor purity. This allowed us to circumvent challenges with incomplete availability of purity estimates across tumor datasets, as well as other limitations of existing methods. For example, expression-based methods like ESTIMATE (Yoshihara et al, Nature Communications 2013) rely on pre-defined contaminating cell signatures, while ABSOLUTE (Carter et al, Nature Biotechnology 2012) uses allele-specific copy numbers and requires user-input to determine ploidy.

Nevertheless, we found that signatures identified and removed using contrastive PCA (cPCA) were highly correlated with existing estimates of tumor purity (Figure 1d). Thus, 'regressing out' the expression signature associated with tumor purity estimates would yield similar results, though it does not account for the multiple tumor-specific expression signatures identified by cPCA.

We also note that in the revised manuscript we have added direct comparisons of our results with those from a related analysis in Yu et al. (Nature Communications, 2019), which used existing estimates of tumor purity to account for the effects of normal cell contamination. This analysis (in revised Supplementary fig 6, section 'Celligner enables improved tumor-cell line similarity estimates', paragraph 2) shows a high level of concordance between our results and theirs (see additional information about the comparison between our method and the Yu et al. method below), further suggesting that both approaches for accounting for variable tumor purity yield broadly similar results.

Nevertheless, we believe an important direction for future work could be to develop flexible methods that can account for available prior estimates of tumor purity, as well as additional 'prior knowledge' such as normal cell expression signatures, while still leveraging the power of unsupervised methods such as cPCA and MNN to best account for the effects of normal cell contamination (revised discussion, paragraph 6).

- Figure b – what is the motivation behind using top 70 PCs, has a sensitivity analysis been done?

We thank the reviewer for highlighting the need for further explanation of this point. We initially selected the number of PCs (70) based on an assessment of their statistical significance using a jackstraw analysis (Chung & Storey, Bioinformatics 2015). However, the resulting estimates of cell line/tumor expression similarity are largely insensitive to the precise choice, and indeed remain quite similar when measuring distances in the full gene space (using data for all 19,188 genes). While performing additional benchmarking analysis for these revisions we determined that the agreement between annotated cell line types and classifications based on a nearest tumor neighbor classifier was actually slightly higher using the full gene space compared to a PCA subspace (see plot below). Given this, and to circumvent the need for introducing an additional free parameter (number of PCs), we have updated our analysis in the revised manuscript to use the full gene space for analysis of tumor/cell line similarity. Note that we still use a PCA subspace to calculate the UMAP embedding and clustering.

Additionally, we have updated our analysis to use Pearson correlation distance (instead of Euclidean distance), as we found this produced similar results overall (see plot below) while providing more directly interpretable estimates of tumor/cell line similarity. These updated methods are reflected in the revised figures Figure 3a, Figure 3b, and Supplementary Figure 4, as well as the nearest neighbor classifications included in Supplementary Table 1.

Agreement between Celligner cell line tumor type classifications and annotated cancer type. We used various PCA subspaces and gene spaces to calculate the distances between cell line and tumors used to classify each cell line. We found that using Pearson correlation distance within the space of all functional genes ($n = 19,188$ genes) produced the highest agreement between the Celligner classifications and the annotated type.

- A comprehensive comparison to the rankings based on simple correlation from Yu et al and Collector would be helpful to evaluate the approach.

We thank the reviewer for pointing out the value of adding more extensive benchmarking comparisons between the results of Celligner and those of previously published methods. In the revised manuscript, we have incorporated extensive comparisons of the tumor/cell line similarity estimates using Celligner with those from a related analysis in Yu et al. (Nature Communications 2019), as well as from CancerCellNet (Peng et al., bioRxiv 2020), a computational method that uses random forest classification to measure the similarity of cancer models' expression profiles with patient tumor data.

Specifically, we compared estimates of how similar each cell line's expression profile is to tumor samples of the same annotated type when using the Celligner-aligned data vs. the Yu et al. method, as well as the random forest classification probabilities from CancerCellNet (see revised Supplementary Figure 6 and Supplementary Figure 7 and the section 'Celligner enables improved tumor-cell line similarity estimates'). Overall, we found good agreement between the Celligner results and the other methods for many of the cancer types.

We also compared the nearest neighbor classifications of cell lines' tumor type based on Celligner with tumor type classifications from Yu et al. and CancerCellNet (see revised methods sections 'Yu et al. comparison' and 'CancerCellNet comparison'). We found that Celligner-based cell line classifications agreed with the annotated cancer type substantially more frequently than those from Yu et al. (Celligner: 50% agreement, Yu et al.: 38%; $n = 666$ cell lines). We also

observed a similar level of improvement with Celligner when compared to CancerCellNet's classifications on a matched cohort (Celligner: 49% agreement; CancerCellNet: 37%; n = 657 cell lines).

Overall these results show that estimates of cell line/tumor transcriptional similarity using Celligner are broadly similar to previously published results for most tumor types, but Celligner shows substantially better agreement when benchmarking tumor-type classification against annotated types.

We were not able to directly compare the results of Celligner to CELLector itself (Najgebauer et al., Cell Systems 2020), as CELLector defines key genomic features and subtypes based on a specific tumor population of interest, and then identifies cell lines that are well represented by those population-specific features. However, in the revised manuscript we have added comparisons of Celligner's expression-based assessment of cell line-tumor similarity with a corresponding analysis based on the genomic features used by CELLector: a dataset of 1,250 clinically relevant mutation, copy number, and methylation features defined by Iorio et al. (Cell 2016) as 'cancer functional events (CFEs)'. We compared the rankings of cell lines by cancer type with these features and using Celligner-aligned expression data (see revised Supplementary Figure 8, section 'Celligner enables improved tumor-cell line similarity estimates' paragraph 3, and methods section 'Cancer Function Event comparison'), and found that, although both datasets classified cell lines as their annotated type with similar accuracy (Celligner: 60% agreement, CFEs: 61% agreement; n = 459 cell lines), the rankings within disease types were dissimilar.

- Have the authors considered other validation approaches?

Although we ultimately lack a 'ground truth' for evaluating the similarity between specific tumors and cell lines, we have incorporated several independent approaches to validate the results of our method, including the additional benchmarking comparisons with previous work added in the revision (as described above).

Beyond showing overall agreement between annotated tumor types and subtypes, we also show that sample populations that we know should be *dissimilar* between the cell lines and tumors are indeed represented distinctly in our Celligner analysis. In our initial manuscript we demonstrated this approach using myeloma samples (Figure 4c/d), which exist only in the cell line data, and kidney chromophobe samples (Figure 4f), which exist only in the tumor data. In the revised manuscript, we have also added an additional example (revised Supplementary Figure 5) using fibroblast cell lines, showing that they cluster distinctly from the tumor samples.

- Integrating mutation data to see if similar cell lines are closer to tumor types of interest based on various omics measurements would be worthwhile.

We thank the reviewer for this suggestion. Some genomic alterations can result in corresponding changes in gene expression (Osmanbeyoglu et al, Nature Communications 2017 & Zhang et al, Nature Genetics 2018) and we did find that some cell lines that had previously been identified as dissimilar to the tumor type in terms of mutations, such as OC316, which

Domcke et al (Nature Communications 2016) identified as being hypermutated, and COLO320, which Ronen et al. (Life Science 2019) found lacked major colorectal driver genes, were also identified as dissimilar to tumors in our analysis. Not all mutations will be reflected in distinct gene expression profiles though, and our systematic comparison between cell line tumor-similarity estimates using Celligner and a set of mutation, copy number, and methylation features showed a large degree of disagreement (see above; revised Supplementary Fig. 8, and section 'Celligner enables improved tumor-cell line similarity estimates' paragraph 3).

Incorporating mutations presents its own challenges as cell lines generally lack paired normal samples, making it difficult to distinguish between cancer-related mutations and germline mutations. Differences in mutation calling pipelines between tumor and cell line data sets can create additional challenges. Overall, we believe that incorporating mutation data, as well as other 'omics' features, in systematic analysis of cell line tumor similarity is an important direction of future work building on Celligner and other methods.

- Are there any clinical or other annotation features of the set of “novel group of transcriptionally and functionally distinct cell lines”? Maybe they are older?

In the revised manuscript we compared the group of 'undifferentiated' cell lines to the remaining cell lines using a number of available annotations, including: year of establishment of the cell line, age of the patient from which the cell line was derived, sex of the patient from which the cell line was derived, and whether the cell lines were derived from primary or metastatic samples (Supplementary Figure 10). Although there is a significant difference ($p=.011$) in the age of the patients from which the cell lines were derived for 'undifferentiated' cell lines (with the undifferentiated cell lines being derived from slightly older patients), the difference was quite small (a difference in means of about 5 years). The difference in the proportion of cell lines derived from primary patient samples (with more of the undifferentiated cell lines being derived from primary samples) is also significant ($p=.027$), although this may be confounded by differences in lineage composition between the two groups. The year of establishment and sex of the cell lines are not significantly different between the two groups of cell lines.

We also examined additional subtype annotations, and observed that the undifferentiated cluster included melanoma tumors (as well as cell lines) that were identified as undifferentiated (Tsoi et al., Cancer Cell 2018) and basal B breast cancer cell lines, which were identified to have more stem-like and mesenchymal expression profiles (Neve et al., Cancer Cell 2006) (Fig. 4b, 'Discussion' paragraph 4).

Minor:

Page 2 typo: tumora should be “tumors”

Thanks for catching that error. It has been corrected.

Reviewer #2:

Overall, this is a well-written and timely paper about the important bioinformatic problem of matching and assessing the relevance of cell lines to the tumor tissues from which they were derived. The authors adapt data harmonization methods commonly used in single-cell RNA-seq for matching cell types across disparate data sets to great effect at the level of gene expression, and even demonstrate tumor subtype-level matching and point out systematic issues with, for example, cell lines derived from CNS tumors. I am enthusiastic about this paper, but have the following suggestions:

We thank the reviewer for these positive remarks.

1) It would be interesting to include a real "negative control" cell line or cell type in this analysis. The authors show important trends like the overall poor alignment of CNS- and thyroid-derived cell lines with tumor tissue expression profiles, but it would be worth knowing how a completely non-neoplastic cell aligns to the tumor tissues. One could even take something like a relatively homogeneous cardiac-derived cell (a tissue where tumors don't really form).

We fully agree with the reviewer about the need for "negative controls" to show how a non-neoplastic cell aligns to the tumor tissues. For the non-neoplastic samples, we chose to use a group of ~40 fibroblast cell lines. While the data for these cell lines was included in our original analysis, we did not highlight their potential use as a negative control. In the revised manuscript (revised Supplementary Figure 5, section 'Alignment preserves meaningful subtype relationships', paragraph 5), we show that nearly all fibroblast cell lines cluster separately from tumor samples, as expected.

2) While the methodology is well-described here, the study lacks a systematic performance comparison to competing methods, which I think should be an essential part of any paper primarily reporting a new technique. The authors mention the work of Yu et al, Nature Communications, 2019 which used Combat and a specific correlation-based strategy to compare cell lines and tumor tissues. While the authors show a UMAP projection of COMBAT results in Supp. Fig. 1, there is a little more to the particular methodology of Yu et al, and there's no real quantitative comparison - just a visualization. In addition, there are other prominent studies that developed very different approaches to this problem. For example, Alvarez et al, Nature Genetics, 2019 also tackles this problem of matching cell lines and other patient-derived models to tumor tissue at the level of gene expression. They use protein activities inferred from RNA-seq data via regulatory network reconstruction. While these are just suggested points of comparison, I feel that some type of quantitative assessment of how this new approach performs relative to others would improve the paper.

We thank the reviewer for pointing out the value of adding more extensive benchmarking comparisons between the results of Celligner and those of previously published methods. In the

revised manuscript, we have incorporated extensive comparisons of the tumor/cell line similarity estimates using Celligner with those from Yu et al. (Nature Communications 2019), as well as from CancerCellNet (Peng et al., bioRxiv 2020), a computational method that uses random forest classification to measure the similarity of cancer models' expression profiles with patient tumor data.

Specifically, we compared estimates of how similar each cell line's expression profile is to tumor samples of the same annotated type when using the Celligner-aligned data vs. the Yu et al. method, as well as the random forest classification probabilities from CancerCellNet (see revised Supplementary Figure 6 and Supplementary Figure 7 and the section 'Celligner enables improved tumor-cell line similarity estimates'). Overall, we found good agreement between the Celligner results and the other methods for many of the cancer types. We also compared the nearest neighbor classifications of cell lines' tumor type based on Celligner with tumor type classifications from Yu et al. and CancerCellNet (see revised methods sections 'Yu et al. comparison' and 'CancerCellNet comparison'). We found that Celligner-based cell line classifications agreed with the annotated cancer type substantially more frequently than those from Yu et al. (Celligner: 50% agreement, Yu et al.: 38%; n = 666 cell lines). We also observed a similar level of improvement with Celligner when compared to CancerCellNet's classifications on a matched cohort (Celligner: 49% agreement; CancerCellNet: 37%; n = 657 cell lines).

Overall these results show that estimates of cell line/tumor transcriptional similarity using Celligner are broadly similar to previously published results for most tumor types, but Celligner shows substantially better agreement when benchmarking tumor-type classification against annotated types.

References

1. Yoshihara, K., Shahmoradgoli, M., Martínez, E., et al. 2013. Inferring tumour purity and stromal and immune cell admixture from expression data. *Nature Communications* 4, p. 2612.
2. Carter, S. L. et al. Absolute quantification of somatic DNA alterations in human cancer. *Nat. Biotechnol.* **30**, 413–421 (2012).
3. Yu, K. et al. Comprehensive transcriptomic analysis of cell lines as models of primary tumors across 22 tumor types. *Nat. Commun.* **10**, 3574 (2019).
4. Chung, N.C. and Storey, J.D. 2015. Statistical significance of variables driving systematic variation in high-dimensional data. *Bioinformatics* 31(4), pp. 545–554.
5. Peng, D. et al. Evaluating the transcriptional fidelity of cancer models. *BioRxiv* (2020). doi:10.1101/2020.03.27.012757
6. Najgebauer, H. et al. CELLector: Genomics Guided Selection of Cancer in vitro Models. *BioRxiv* (2018). doi:10.1101/275032
7. Iorio, F. et al. A landscape of pharmacogenomic interactions in cancer. *Cell* **166**, 740–754 (2016).
8. Osmanbeyoglu, H.U., Toska, E., Chan, C., Baselga, J. and Leslie, C.S. 2017. Pancancer modelling predicts the context-specific impact of somatic mutations on transcriptional programs. *Nature Communications* 8, p. 14249.

9. Zhang, W., Bojorquez-Gomez, A., Velez, D.O., et al. 2018. A global transcriptional network connecting noncoding mutations to changes in tumor gene expression. *Nature Genetics* 50(4), pp. 613–620.
10. Domcke, S., Sinha, R., Levine, D. A., Sander, C. & Schultz, N. Evaluating cell lines as tumour models by comparison of genomic profiles. *Nat. Commun.* **4**, 2126 (2013).
11. Ronen, J., Hayat, S. & Akalin, A. Evaluation of colorectal cancer subtypes and cell lines using deep learning. *Life Sci. Alliance* **2**, (2019).
12. Tsoi, J. *et al.* Multi-stage Differentiation Defines Melanoma Subtypes with Differential Vulnerability to Drug-Induced Iron-Dependent Oxidative Stress. *Cancer Cell* **33**, 890–904.e5 (2018).
13. Neve, R. M. *et al.* A collection of breast cancer cell lines for the study of functionally distinct cancer subtypes. *Cancer Cell* **10**, 515–527 (2006).

Reviewers' Comments:

Reviewer #1:

Remarks to the Author:

The authors have adequately addressed my comments and suggestions.

Reviewer #2:

Remarks to the Author:

In my opinion, the authors have done a thorough job of addressing my previous criticisms.

REVIEWERS' COMMENTS

Reviewer #1 (Remarks to the Author):

The authors have adequately addressed my comments and suggestions.

Thank you for your review.

Reviewer #2 (Remarks to the Author):

In my opinion, the authors have done a thorough job of addressing my previous criticisms.

Thank you for your review.